# Feasibility of combining short tandem repeats (STRs) haplotyping with preimplantation genetic diagnosis (PGD) in screening for beta thalassemia

Vu Viet Ha Vuong[1,2�], Thinh Huy Tran[1,3,4�‡], Phuoc-Dung Nguyen[1‡], Nha Nguyen Thi[2‡], Phuong Le Thi[1‡], Dang Thi Minh Nguyet[1‡], Manh-Ha Nguyen[4‡], The-Hung Bui[1,5‡], Thanh Van Ta[1,3,4‡], Van-Khanh Tran[1] *

1 Center for Gene and Protein Research, Hanoi Medical University, Hanoi, Vietnam, 2 Hospital of Post and Telecommunications, Hanoi, Vietnam, 3 Biochemistry Department, Hanoi Medical University, Hanoi, Vietnam, 4 Hanoi Medical University Hospital, Hanoi Medical University, Hanoi, Vietnam, 5 Center for Molecular Medicine, Clinical Genetics Unit, Karolinska Institutet, Karolinska University Hospital, Stockholm, Sweden

ो These authors contributed equally to this work.
‡ THT, PDN, NNT, PLT, DTMN, MHN, THB and TVT also contributed equally to this work.
* tranvankhanh@hmu.edu.vn

**Data Availability Statement:** All relevant data are within the paper and its Supporting information files.

## Abstract

β-thalassemia is an autosomal recessive disease with the reduction or absence in the production of β-globin chain in the hemoglobin, which is caused by mutations in the Hemoglobin subunit beta (*HBB*) gene. In Vietnam, the number of β-thalassemia carriers range from 1.5 to 25.0%, depending on ethnic and geographical areas, which is much higher than WHO's data worldwide (1.5%). Hence, preimplantation genetic diagnosis (PGD) plays a crucial role in reducing the rate of β-thalassemia affected patients/carriers. In this research, we report the feasibility and reliability of conducting PGD in combination with the use of short tandem repeat (STR) markers in facilitating the birth of healthy children. Six STRs, which were reported to closely linked with the *HBB* gene, were used on 15 couples of β-thalassemia carriers. With 231 embryos, 168 blastocysts were formed (formation rate of 72.73%), and 88 were biopsied and examined with STRs haplotyping and pedigree analysis. Thus, the results were verified by Sanger sequencing, as a definitive diagnosis. Consequently, 11 over 15 couples have achieved pregnancy of healthy or at least asymptomatic offspring. Only three couples failed to detect any signs of pregnancy such as increased Human Chorionic Gonadotropin (HCG) level, foetal sac, or heart; and one couple has not reached embryo transfer as they were proposed to continue with HLA-matching to screen for a potential umbilical cord blood donor sibling. Thus, these results have indicated that the combination of PGD with STRs analysis confirmed by Sanger sequencing has demonstrated to be a well-grounded and practical clinical strategy to improve the detection of β-thalassemia in the pregnancies of couples at-risk before embryo transfer, thus reducing β-thalassemia rate in the population.

**Funding:** The author(s) received no specific funding for this work.

**Competing interests:** The authors have declared that no competing interests exist.

## Introduction

β-thalassemia is an autosomal recessive disease with the reduction or absence in the production of β-globin chain, which is caused by mutations in the *HBB* gene. Based on the zygosity of the β-thalassemia mutations, it is categorized into three groups: a heterozygous mutation results in a β-thalassemia carriers, expressing mild symptoms; and the two others carrying homozygous mutations are β-thalassemia major and intermedia, depending on the severity of the mutations.

According to WHO, about 1.5% of the world population are β-thalassemia carriers [1], in which 0.5 to 12.8% belongs to South-East Asian countries [2]. In Vietnam, this number is even higher, from 1.5 to 25.0%, depending on ethnic and geographical areas [3–5].

One notable difference between α- and β-thalassemia is that not only until the first 6 months of life, β-thalassemia patients express anemia as the HbF production declines and the presence of HbB is required [6]. Hence, preimplantation genetic diagnosis (PGD) is considered as an important tool in cutting down the rate of β-thalassemia infants, which would reduce the financial burden on families of β-thalassemia carriers. Additionally, PGD in conjunction with *in vitro* fertilisation facilitate the birth of a healthy child by selecting the genetically non-affected embryos to transfer, therefore, no abortion is required. Hence, it would be more popularly acceptable than prenatal genetic diagnosis.

Various indirect and direct strategies using PCR-based techniques have been proposed to optimise the PGD procedure of β-thalassemia, such as allele-specific reverse dot blot, single-stranded confirmation polymorphism and denaturing gradient gel electrophoresis [7], or nested PCR and direct sequencing [8]. However, there are more than 200 of known *HBB* mutations [9], the risk of having a false negative is quite high due to the possible errors in experiment design or allele drop out due to the low DNA content.

Recently, the use of microsatellite markers such as short tandem repeats (STRs) have gained prominent potentials in PGD for β-thalassemia as these selected STRs are linked closely with the *HBB* gene which helps provide linkage analysis of the mutant presences in the family and avoid allele drop out. Additionally, a series of STRs specific for Vietnamese population has been developed [10], which would further increase the precision of the method.

In this report, we presented the retrospective review of the clinical utility of STRs haplotyping in PGD to facilitate the birth of healthy/asymptomatic children of β-thalassemia carrier couples received reproductive service at our centre.

## Methods

### Ethics statement

The study was approved by Hanoi Medical University Institutional Ethical Review Board, ID 470/GCN-HDDDNCYSH-DHYHN. Participants were informed, and their written consents were documented after receiving genetic counselling from clinical geneticist.

### Patients

This is a retrospective study of patients, including fifteen couples at risks of having β-thalassemia offspring requested for *in vitro* fertilization (IVF) procedures and PGD at Post Hospital during 2020–2021. Their sequencing results indicated that they were β-thalassemia carriers (heterozygotes—possessing one defective alleles). They all received the similar controlled ovarian stimulation, Intracytoplasmic Sperm Injection (ICSI) protocol and preimplantation genetic screening (PGS).

## Test procedure

Each couple and additional family members' blood samples were subjected to hemoglobin electrophoresis, β-thalassemia gene mutation detection and STRs linkage analysis. DNA samples of the couples and family members were extracted from peripheral blood using QIAmp® DNA Mini Kit (Ref 51304).

To obtain the DNA profile, 6 STR markers were used, 4 upstream (D11S1243, HBB5138, HBB5178, HBB5205) and 2 downstream (D11S1760, HBB5576), which are closely linked with the *Hbb* gene (Table 1). STRs primers were obtained accordingly to the literature [11]. Multiplex PCR was performed to amplify the STR markers by QIAGEN Multiplex PCR Kit (100) (ID: 206143), and the fragments were separated using Applied Biosystem Bioanalyser 3500. The data was observed and analysed on GeneMarker software.

Embryo cell samples were received from IVF centre after embryo biopsy at day 5. Hence, whole genome amplification (WGA) was performed using QIAGEN REPLI® Single Cell (Ref 150343). STR haplotyping was conducted for the DNA profiles comparison with the couples and family members' profiles to screen for the mutated alleles.

## Results

### Pedigree analysis

Couples' characteristics were represented in Table 2. For most couples, as linkage analysis only provides whether the presence of the mutated allele received parentally, it lacks evidence to confirm which allele carrying β-thalassemia mutation for couples with no β-thalassemia children. Hence, samples from couples' parents were required to conduct the examination.

For *couple 15*, as they had one child carrying the HBB mutations, thus, only samples from the parents and the child were required for STR haplotyping, however, samples of additional family members were still used to increase the certainty of the method. The STR haplotyping results of *couple 15* (Fig 1A) were used to obtain the STRs profile of the family. Hence, in alignment with the Sanger sequencing results of each family members, the mutated alleles were profiled and tracked in the embryos *via* pedigree analysis. As observed in *Couple 15*, the mutated alleles were detected and transferred from the parents to the child and the embryo HN4 and HN5 carrying CD41/42 and CD17 respectively (Fig 1B), which agreed with the Sanger sequencing results (S1 Fig).

**Table 1. STR markers properties.**

| | STR markers | Repeat motif | Size | Primer 5'–3' (F & R) |
|---|---|---|---|---|
| Upstream | D11S1243 | (TG)n | 202–238 | **HEX** -CTGCCCTAATTCTGTCTACC |
| | | | | GTTGTGCACCATGAAGATACAC |
| | HBB5138 | (AC)n | 386–410 | **HEX** -AGAAATGTCCTTTAGAGAAATACCTTC |
| | | | | GTGGAGAGGAATCTGTTACTG |
| | HBB5178 | (TG)n | 137–171 | **FAM** -CGTAATTGCTTTCAGTACCATTTATG |
| | | | | GATGTATTCGTCAACAGATAAATGG |
| | HBB5205 | (AGAT)n | 380–428 | **FAM** -CCAGGGTAGGTGACATATAC |
| | | | | GTAACTCAAAAAATGGGACCCAAAC |
| Down-stream | D11S1760 | (CA)n | 174–220 | **FAM** -ACCCTGAGTGTCTTCAAAACTC |
| | | | | CAATACTGCTGCATCATGACT |
| | HBB5576 | (AAGG)n | 306–348 | **FAM** -TCCTTCAGGTAAGAAGGAGC |
| | | | | CTTGAAGAGGCTAGGTGC |

**Table 2. Characteristics of couples.**

| Couple | Age, female/male (years) | Female genotype | Male genotype | Pregnancy history |
|---|---|---|---|---|
| 1 | 29/30 | CD35 | CD26 | - |
| 2 | 28/28 | CD17 | CD41/42 | 1 boy after the first PGD cycle |
| 3 | 31/30 | CD17 | CD26 | - |
| 4 | 25/36 | CD17 | CD26 | - |
| 5 | 26/28 | CD26 | CD26 | No pregnancy after the first embryo cycle |
| 6 | 27/29 | CD17 | CD41/42 | - |
| 7 | 33/34 | CD41/42 | CD41/42 | - |
| 8 | 32/32 | CD17 | CD17 | - |
| 9 | 29/30 | CD26 | CD71/72 | - |
| 10 | 26/31 | CD41/42 | CD26 | No pregnancy after the first embryo cycle |
| 11 | 33/35 | CD41/42 | CD41/42 | - |
| 12 | 38/39 | CD26 | CD26 | - |
| 13 | 28/28 | CD27/28 | CD41/42 | - |
| 14 | 26/26 | CD26 | CD41/42 | - |
| 15 | 28/28 | CD17 | CD41/42 | Has one child carrying CD17 and CD41/42 mutations |

The STR markers demonstrated high persistence in the results with only the embryo HN3 showing a different haplotype from the parents, as well as the missing of the HBB5138 and HBB5178 signals. This may be due to some problem in the WGA process, occurring a complete locus drop-out at HBB5138 and an allelic drop-out in HBB5178, as the QIAGEN REPLI® Single Cell was used for WGA still has a small percentage of ADO [12].

## Blastocysts

The data regarding the blastocysts from 16 couples is presented in Table 3. Generally, with 271 oocytes underwent ICSI, 229 formed embryos, and only 173 developed into blastocysts, leading to a blastocyst formation rate of 75.55%. Hence, blastocysts were biopsied and followed by WGA and STRs analysis with Sanger sequencing simultaneously to confirm the results for potentially transferred embryos.

## PGD outcome

Among 168 formed blastocysts, 88 blastocysts were examined for the carrying of mutated alleles received parentally, which resulted in 18 normal, 35 carriers, and 18 affected blastocysts, contributing to the transferable embryos of 52. However, there were 14 embryos carried chromosomal abnormalities (detected by PGS) and only 3 out of 88 failed to be amplified, thus, continue with the diagnosis. Hence, the rate of blastocyst amplification is 96.59%. Overall, 11 over 15 couples achieved pregnancy, three couples with no signs of successful pregnancy (Beta HCG, foetal sac, or foetal heart), one with the unsuccessful pregnancy during the first PGD cycle but achieved the second cycle and the last couple with a β-thalassemia major first born have yet proceeded to embryo transfer (Table 4).

For couple 15, they had a β-thalassemia major first born, they were advised to continue with human leukocyte antigen (HLA) matching to facilitate the birth of an HLA-matched healthy sibling, who can provide hematopoietic stem cells (HSC) transplant for the first born.

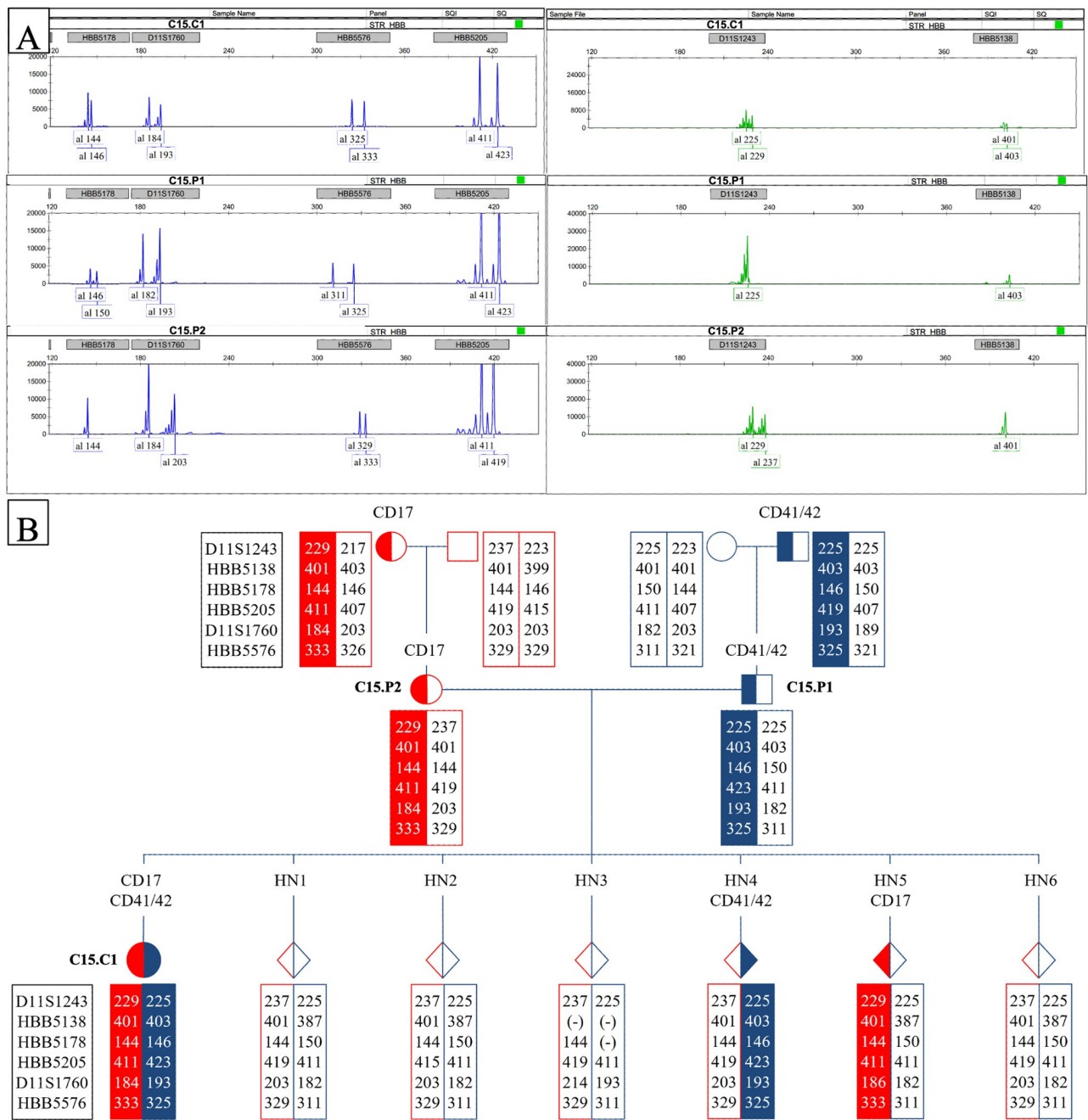

**Fig 1. STRs data and pedigree analysis.** (A) STRs data analysed on GeneMarker Software, with HBB5178, D11S1760, HBB5576, HBB5205 in blue, and D11S1243, HBB5138 in green; (B) Pedigree analysis of Couple 15, with mutated allele received from the mother (in red) and the father in (blue). The HBB5138 and HBB5178 signals were not detected for HN3, thus, marked as (-).

## Discussion

β-thalassemia is a popular genetic disease with a high percentage of carriers, especially in Vietnam, with proportions vary among ethnical and geographical groups such as Tay and Muong with high percentage of carriers (10.7 and 11%, respectively) [13]. On top of that, the movement of different ethnicities to metropolises for occupations and settlements further increases

**Table 3. Summary of the blastocyst development.**

| Couple | Oocytes MII | Oocytes ICSI | Embryos | Blastocysts | Blastocyst formation rate % | Biopsied blastocysts | Blastocyst amplified | Amplified rate % |
|---|---|---|---|---|---|---|---|---|
| 1 | 14 | 13 | 12 | 6 | 50.0 | 3 | 3 | 100 |
| 2 | 37 | 36 | 28 | 16 | 57.14 | 7 | 7 | 100 |
| 3 | 15 | 14 | 13 | 10 | 76.92 | 6 | 6 | 100 |
| 4 | 11 | 11 | 10 | 6 | 60.0 | 5 | 5 | 100 |
| 5 | 14 | 14 | 14 | 7 | 50.0 | 7 | 7 | 100 |
|  | 19 | 16 | 15 | 10 | 66.67 | 10 | 10 | 100 |
| 6 | 13 | 12 | 12 | 12 | 100 | 5 | 5 | 100 |
| 7 | 20 | 18 | 15 | 13 | 86.67 | 4 | 4 | 100 |
| 8 | 20 | 20 | 17 | 17 | 100.0 | 7 | 7 | 100 |
| 9 | 24 | 24 | 21 | 19 | 90.48 | 8 | 8 | 100 |
| 10 | 12 | 11 | 11 | 11 | 100.0 | 7 | 7 | 100 |
| 11 | 12 | 12 | 6 | 6 | 100.0 | 4 | 4 | 100 |
| 12 | 15 | 15 | 14 | 7 | 50.0 | 3 | 3 | 100 |
| 13 | 24 | 24 | 21 | 20 | 95.24 | 4 | 2 | 50 |
| 14 | 22 | 22 | 11 | 2 | 18.18 | 2 | 2 | 100 |
| 15 | 11 | 11 | 11 | 6 | 54.54 | 6 | 6 | 100 |
| SUM | 283 | 273 | 231 | 168 | 72.73 | 88 | 85 | 96.59 |

the β-thalassemia population due to cross-ethnic marriage. Thus, despite the attempt of expanding blood transfusion network between national to provincial hospitals to treat β-thalassemia patients, it is being exhausted due to the increasing rate of β-thalassemia infants [13]. This problem raises both financial and social burden to the patients' family as well as the

**Table 4. Summary of PGD outcomes.**

| Couple | Biopsied blastocysts | Chromosomal abnormalities | Normal | Carriers | Affected | Transferred embryo genotype | Beta hCG | Fetal sacs | Fetal heart | Outcome |
|---|---|---|---|---|---|---|---|---|---|---|
| 1 | 3 |  |  | 3 |  | $\beta^N/\beta^{CD26}$ | x |  |  | No pregnancy after the first PGD cycle |
| 2 | 7 | 2 | 1 | 4 |  | $\beta^N/\beta^N$ | x | x | x | 1 girl after the second PGD cycle |
| 3 | 6 | 4 |  | 2 |  | $\beta^N/\beta^{CD26}$ | x | x | x | 5 months pregnancy |
| 4 | 5 |  | 2 | 3 |  | $\beta^N/\beta^N$ | x | x | x | Pregnancy |
| 5 | 7 | 3 | 1 | 1 | 2 | $\beta^N/\beta^N$ | x | x |  | 1 boy after the second embryo transfer |
|  | 10 | 3 | 2 | 3 | 2 | $\beta^N/\beta^N$ | x | x | x |  |
| 6 | 5 |  | 2 |  | 3 | $\beta^N/\beta^N$ | x | x | x | 1 boy |
| 7 | 4 |  |  | 2 | 2 | $\beta^N/\beta^{CD41/42}$ | x | x | x | 1 boy |
| 8 | 7 |  | 4 |  | 3 | $\beta^N/\beta^N$ | x | x | x | 1 boy |
| 9 | 8 | 1 | 2 | 3 | 2 | $\beta^N/\beta^N$ | x | x | x | 1 boy |
| 10 | 7 | 1 |  | 4 | 2 | $\beta^N/\beta^{CD26}$ |  |  |  | No pregnancy after 2 embryo transfers |
|  |  |  |  |  |  | $\beta^N/\beta^{CD26}$ |  |  |  |  |
| 11 | 4 |  |  | 3 | 1 | $\beta^N/\beta^{CD41/42}$ | x | x | x | 1 girl |
| 12 | 3 |  |  | 2 | 1 | $\beta^N/\beta^{CD26}$ |  |  |  | No pregnancy after the first PGD cycle |
| 13 | 4 |  |  | 2 |  | $\beta^N/\beta^{CD27/28}$ | x | x | x | 1 boy |
| 14 | 2 |  | 1 | 1 |  | $\beta^N/\beta^N$ | x | x | x | Pregnancy |
| 15 | 6 |  | 3 | 2 |  |  | Transferred embryo was not yet decided ||||
| SUM | 88 | 14 | 18 | 35 | 18 |  |  |  |  |  |

government, which, to a greater extent, stresses the importance of genetic examination, especially, PGD in combination with IVF.

PGD coping with traditional methods, for example, PCR, sequencing may experience risk of complete locus or allelic dropout (from 10 to 25%) due to the inadequate DNA content [14, 15], which may cause misdiagnoses. The application of multiple displacement amplification, which is currently commercialised in WGA kits such as QIAGEN REPLI$^{®}$ Single Cell, has greatly aided as it increases the genetic material obtained from embryo biopsies with the successful amplification rate of 96.59%, comparable to the reported rate of previous report using the same method (98%) [12]. Hence, STR haplotyping was applied to track the mutated alleles transferred in the family. The method has also proven credibility in other monogenic disorders such as Duchene muscular dystrophy, haemophilia [16, 17]. The selection of markers is the crucial key to the precision of the results in this method as the informativeness of the markers was based on the STR heterozygosity and their tight association to the causative gene. The six selected STRs surround and link closely to the HBB gene (within 0.7 mb) and demonstrated high polymorphic information content and expected heterozygosity in previous research in both Vietnamese as well as Asian populations [10, 11]. Thus, the STR loci was amplified along with the pathogenic loci simultaneously which can maximally monitor and minimize the ADO rate. The potential embryos selected *via* STR haplotyping were confirmed with Sanger sequencing, of which results stay consistent in most cases (55 over 56 transferable embryos ~98.21%), except for the HN3 of couple 15 with much noise data even after purified with Exo-SAP-IT™. This might be due to the contamination during WGA or sample handling afterwards. Additionally, the time conducting STR analysis, and the price are relatively lower than using conventional methods for known mutations. Hence, indirect methods such as STR analysis may pave the way for a better PGD procedure with more ease of use and less resources consumption, especially in β-thalassemia.

Despite the fact that our research used less STR markers than the previous research (6 compared to 15) and confirmed the results by Sanger sequencing, no problem regarding the heterozygosity affecting the detection of allele was found, thus, proven the effectiveness of this strategy. Additionally, other method such as reverse dot blot can be used to cross check the results [11], however, by confirming with Sanger sequencing, a wider spectrum of *HBB* mutations can be checked. Currently, in developed countries, the application of next generation sequencing (NGS) in NGS-based SNP haplotyping has proven superiority as it can reduce misdiagnosis by linkage analyses and detect aneuploidy simultaneously [18]. Combining PGS and PGD to exclude chromosomal abnormal embryo had shown a statistically increased pregnancy rate and 3-fold reduction of spontaneous abortion rate compared with PGD alone [19]. Thus, the ability of detect both mutations and chromosomal abnormalities in one protocol greatly reduce labour works and potential mishandling. However, due to the shortage of availability and expensiveness of NGS, it cannot serve the lower-income populations, therefore, cannot achieve the purpose of increase the accessibility to PGD for β-thalassemia in developing countries.

In this project, we facilitated the birth/pregnancy of 11 out of 15 couples, in which seven couples carried completely normal children and four were carriers, making the clinical pregnancy rate of 73.33% higher than the standard rate of 62% [20], though after implantation 12 over 16 embryos developed into foetal sacs, making the implantation rate of 75%. Positively, nine couples achieved pregnancy after first PGD cycle, thus, the rate of pregnancy after first PGD cycle of 60%, which are relatively higher than previous research [21, 22]. However, the research sample number was still small and inadequate to raise any significant conclusion regarding the method. We also used day 5 or 6 embryos, which have developed into blastocyst stage for biopsy, evaluation of the embryo growth, and transfer, hence, achieved higher implantation

rate and pregnancy/live-birth rate compared to day-3 embryos, as well as increase the chance of detecting the presence of mosaicism due to the ability of biopsy higher number of cells [23, 24]. The vitrification system was also highly important as we can store the freeze blastocysts during the time waiting for genetic examination to select the best embryos for the transfer. This strategy has been proven to reduce the likelihood of ovarian hyperstimulation syndrome [25, 26]. which possibly leads to life-threatening in some severe situations.

As HLA gene display a spectacular degree of polymorphism [27], and the ideal donor for HSC transplant should be compatible at HLA-A, HLA-B, HLA-C and DBR1 [28]. Thus, if it is possible to implement both PGD and HLA-matching to have a normal child and simultaneously find a permanent cure for the first one should be highly recommended. This has been recommended to the couple with one infected offspring.

## Conclusion

In this report, by the use of an indirect method, STR markers to track the mutated alleles transmitting in the family, we have facilitated the birth of nine babies, three pregnancies, all were healthy or asymptomatic, carrying only one mutated allele of HBB gene. Only three couples resulted in no pregnancy with no increase in Beta HCG level or detection of the foetal sac or heart *via* ultrasound. One couple have not reached embryo transfer due to as they were waiting for the HLA typing results. Thus, STR haplotyping is a much cheaper method for the detection of mutated alleles running in a family, which has proved to exert reliable results.

## Supporting information

**S1 Fig. Sequencing data of *couple 15* and the transferable embryos: C15.P1 & C15.P2 were the parents, and the others were the embryos (HN1 to HN6), indicating similar results to ones using STR haplotyping.**
(TIFF)

## Acknowledgments

We thank the patients and their families for their voluntary involvement in this study.

## Author Contributions

**Conceptualization:** Thanh Van Ta, Van-Khanh Tran.

**Data curation:** Vu Viet Ha Vuong, Thinh Huy Tran, Phuoc-Dung Nguyen, Phuong Le Thi, Manh-Ha Nguyen.

**Formal analysis:** Vu Viet Ha Vuong, Thinh Huy Tran, Phuoc-Dung Nguyen, Nha Nguyen Thi, Phuong Le Thi.

**Investigation:** Nha Nguyen Thi, Phuong Le Thi, Manh-Ha Nguyen.

**Methodology:** Vu Viet Ha Vuong, Thinh Huy Tran, Phuoc-Dung Nguyen, Nha Nguyen Thi, Phuong Le Thi, Manh-Ha Nguyen.

**Supervision:** Thanh Van Ta, Van-Khanh Tran.

**Validation:** Dang Thi Minh Nguyet, The-Hung Bui, Thanh Van Ta.

**Visualization:** Vu Viet Ha Vuong, Thinh Huy Tran, Phuoc-Dung Nguyen.

**Writing – original draft:** Vu Viet Ha Vuong, Thinh Huy Tran, Phuoc-Dung Nguyen.

**Writing – review & editing:** Dang Thi Minh Nguyet, The-Hung Bui, Thanh Van Ta, Van-Khanh Tran.

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
