## [Decision Letter · Decision Letter 0]

8 Jun 2022

PONE-D-22-11235Feasibility of combining short tandem repeats (STRs) haplotyping with preimplantation genetic diagnosis (PGD) in screening for beta thalassemiaPLOS ONE

Dear Dr. Tran,

Thank you for submitting your manuscript to PLOS ONE. After careful consideration, we feel that it has merit but does not fully meet PLOS ONE’s publication criteria as it currently stands. Therefore, we invite you to submit a revised version of the manuscript that addresses the points raised during the review process.

We look forward to receiving your revised manuscript.

Kind regards,

J Francis Borgio, Ph.D.,

Academic Editor

PLOS ONE

Journal Requirements:

3. We note you have included a table to which you do not refer in the text of your manuscript. Please ensure that you refer to Table 4 in your text; if accepted, production will need this reference to link the reader to the Table.

Additional Editor Comments:

Authors shall consider all the suggestions carefully from the reviewers

Reviewers' comments:

Reviewer's Responses to Questions

**Comments to the Author**

1. Is the manuscript technically sound, and do the data support the conclusions?

Reviewer #1: Partly

Reviewer #2: Yes

2. Has the statistical analysis been performed appropriately and rigorously? 

Reviewer #1: Yes

Reviewer #2: Yes

3. Have the authors made all data underlying the findings in their manuscript fully available?

Reviewer #1: No

Reviewer #2: Yes

4. Is the manuscript presented in an intelligible fashion and written in standard English?

Reviewer #1: Yes

Reviewer #2: Yes

5. Review Comments to the Author

Reviewer #1: • The topic of this article lacks new ideas: PGD is a mature technology in the current field of reproductive medicine, and it is also a conventional indication for beta thalassemia. What is the novelty of this paper?

• Two STR markers named HBB5178, and D11S1760 have approximately close size range which can make misdiagnosis. It would be better to tag them with different fluorescent labels

• There are some grammatical. For instance:

o Line119: " …family members were still used for to increase the…" should change into "to increase"

o Line 151: "…3 over 88 failed…" should change into 3 out of 88 failed

o Line 195: "…11 over 15 couples" should change into 11 out of 15 couples

• The discussion part lack the description of the experience from other research.

• The figures are blurred, it must be changed into a clear one.

Reviewer #2: Over all it is good written article. I have some suggestions to improve it a little bit before publication.

1- If you are using some abbreviations, please give the complete name or description at the first place it is used in the article. Like use of HBB, HCG etc.

2- The line 166-168 sentence is confusing. How the movement of different ethnicities to metropolises does increase beta thalassemia in the population?

3- In line 210 add "is" after "thus, if it"

6. PLOS authors have the option to publish the peer review history of their article (what does this mean?). If published, this will include your full peer review and any attached files.

Reviewer #1: No

Reviewer #2: **Yes: **Muhammad Farooq Sabar

---

## [Author Response · Author response to Decision Letter 0]

10 Oct 2022

RESPOND TO REVIEWER

Submission Title: Feasibility of combining short tandem repeats (STRs) haplotyping with preimplantation genetic diagnosis (PGD) in screening for beta thalassemia

Dear Editors and Reviewers, 

We are grateful for your time reading and considering our work. Your feedbacks have helped us improved our manuscript substantially. We have thoroughly revised our work and, to the best of our ability as non-native speaker correct the grammar and syntax to meet with the publication standard. We hereby submit our revised manuscript together with detailed respond to each of the queries raised by the reviewer.

Respond to the Reviewer 1’s queries

Q: The topic of this article lacks new ideas: PGD is a mature technology in the current field of reproductive medicine, and it is also a conventional indication for beta thalassemia. What is the novelty of this paper?

Answer:

It is true that PGD is a mature technology in the current field of reproductive medicine and a conventional indication for beta thalassemia. Although PGD was first reported in the world in 1988, not until 2015, in a technology transfer project by Military Medical University, PGD was first introduced in Vietnam. Hence, with only 7 years of technological adaptation and optimization for Vietnamese population, there were many aspects that we can adjust and research on. Furthermore, not many hospitals in Vietnam have the adequate technological requirements as well as received the procedure of PGD for beta thalassemia. In this research, we would like to introduce the feasibility and reliability of this PGD strategy for beta thalassemia to other facilities in Vietnam and other developing countries with later obtention of the technology, thus, to reduce the rate of thalassemia carriers. 

Q: Two STR markers named HBB5178, and D11S1760 have approximately close size range which can make misdiagnosis. It would be better to tag them with different fluorescent labels

Answer:

We will definitely change the fluorescence colour for the markers HBB5178, and D11S1760 in the next experiments. In this research, due to data in previous research, as well as our experience, the two markers are distinguishable with no overlaps. Furthermore, we used other markers to cross-check the results as at least three markers would confirm the presence of the allele. 

Q: There are some grammatical errors

Answer: We have made appropriate corrections of the errors.

Q: The discussion part lacks the description of the experience from other research.

Answer: We have added some description of experience from studies in the same field.

Q: The figures are blurred, it must be changed into a clear one.

Answer: We have changed the figure and increase the resolution to 300 dpi.

Respond to the Reviewer 2’s queries

Q: If you are using some abbreviations, please give the complete name or description at the first place it is used in the article.

Answer: We have added the complete name/description before used abbreviations.

Q: How the movement of different ethnicities to metropolises does increase beta thalassemia in the population?

Answer: We have rephrased the sentence for better understanding. Some ethnicities in Vietnam were found with high frequency of beta-thalassemia mutated allele, therefore, as they moved and settled in metropolises, the rate of carriers and mutated allele frequency in the new population would increase.

Q: Some grammatical errors

Answer: We have made appropriate corrections of the errors.

---

## [Decision Letter · Decision Letter 1]

18 Nov 2022

Feasibility of combining short tandem repeats (STRs) haplotyping with preimplantation genetic diagnosis (PGD) in screening for beta thalassemia

PONE-D-22-11235R1

Dear Dr. Tran,

We’re pleased to inform you that your manuscript has been judged scientifically suitable for publication and will be formally accepted for publication once it meets all outstanding technical requirements.

Kind regards,

J Francis Borgio, Ph.D.,

Academic Editor

PLOS ONE

Reviewers' comments:

Reviewer's Responses to Questions

**Comments to the Author**

1. If the authors have adequately addressed your comments raised in a previous round of review and you feel that this manuscript is now acceptable for publication, you may indicate that here to bypass the “Comments to the Author” section, enter your conflict of interest statement in the “Confidential to Editor” section, and submit your "Accept" recommendation.

Reviewer #2: All comments have been addressed

Reviewer #3: All comments have been addressed

2. Is the manuscript technically sound, and do the data support the conclusions?

Reviewer #2: (No Response)

Reviewer #3: Yes

3. Has the statistical analysis been performed appropriately and rigorously? 

Reviewer #2: (No Response)

Reviewer #3: Yes

4. Have the authors made all data underlying the findings in their manuscript fully available?

Reviewer #2: (No Response)

Reviewer #3: (No Response)

5. Is the manuscript presented in an intelligible fashion and written in standard English?

Reviewer #2: (No Response)

Reviewer #3: Yes

6. Review Comments to the Author

Reviewer #2: (No Response)

Reviewer #3: (No Response)

7. PLOS authors have the option to publish the peer review history of their article (what does this mean?). If published, this will include your full peer review and any attached files.

Reviewer #2: **Yes: **Muhammad Farooq Sabar

Reviewer #3: No

---

## [Editor Report · Acceptance letter]

29 Nov 2022

PONE-D-22-11235R1 

Feasibility of combining short tandem repeats (STRs) haplotyping with preimplantation genetic diagnosis (PGD) in screening for beta thalassemia 

Dear Dr. Tran:

I'm pleased to inform you that your manuscript has been deemed suitable for publication in PLOS ONE. Congratulations! Your manuscript is now with our production department. 

Kind regards, 

on behalf of

Dr. J Francis Borgio 

Academic Editor

PLOS ONE